# The Role of Epicardial Adipose Tissue-Derived MicroRNAs in the Regulation of Cardiovascular Disease: A Narrative Review

**DOI:** 10.3390/biology12040498

**Published:** 2023-03-25

**Authors:** Il-Kwon Kim, Byeong-Wook Song, Soyeon Lim, Sang-Woo Kim, Seahyoung Lee

**Affiliations:** 1Institute for Bio-Medical Convergence, College of Medicine, Catholic Kwandong University, Gangneung-si 25601, Republic of Korea; 2International St. Mary’s Hospital, Catholic Kwandong University, Incheon 22711, Republic of Korea

**Keywords:** epicardial adipose tissue, miRNA, cardiovascular disease

## Abstract

**Simple Summary:**

Cardiovascular diseases are the leading cause of death. Obesity has been associated with multiple pathologic cardiovascular-related conditions. Epicardial adipose tissue is likely the principal adipose tissue relevant to the development of cardiovascular disease due to its close location to the myocardium. Since miRNAs have been implicated in various diseases, including cardiovascular diseases, it is likely that the pathophysiology of epicardial adipose tissue in cardiovascular disease is also regulated by miRNAs. In the present review, we summarized miRNAs reported to be differentially expressed in human epicardial adipose tissue under pathologic conditions, focusing on their potential targets and their roles in the regulation of epicardial adipose tissue biology. As discussed in this review, it is clear that epicardial adipose tissue-derived miRNAs can make a vital contribution to the cardiovascular system both locally and systemically. However, much of the underlying mechanisms with which epicardial adipose tissue-derived miRNAs regulate both the physiology and pathophysiology of the cardiovascular system still largely remain speculative, requiring further studies.

**Abstract:**

Cardiovascular diseases have been leading cause of death worldwide for many decades, and obesity has been acknowledged as a risk factor for cardiovascular diseases. In the present review, human epicardial adipose tissue-derived miRNAs reported to be differentially expressed under pathologic conditions are discussed and summarized. The results of the literature review indicate that some of the epicardial adipose tissue-derived miRNAs are believed to be cardioprotective, while some others show quite the opposite effects depending on the underlying pathologic conditions. Furthermore, they suggest that that the epicardial adipose tissue-derived miRNAs have great potential as both a diagnostic and therapeutic modality. Nevertheless, mainly due to highly limited availability of human samples, it is very difficult to make any generalized claims on a given miRNA in terms of its overall impact on the cardiovascular system. Therefore, further functional investigation of a given miRNA including, but not limited to, the study of its dose effect, off-target effects, and potential toxicity is required. We hope that this review can provide novel insights to transform our current knowledge on epicardial adipose tissue-derived miRNAs into clinically viable therapeutic strategies for preventing and treating cardiovascular diseases.

## 1. Introduction

Cardiovascular diseases (CVDs) have been the leading cause of death worldwide for many decades, and the number of CVD-related deaths reached 18.6 million in 2019 [1]. The risk factors for CVD include hypertension, high blood cholesterol levels, stress, smoking, diabetes, and obesity [2]. Among these factors, obesity, not just being a well-recognized independent risk factor to CVD-related death, has been one of the most critical global health care problems [3]. Obesity also has been associated with multiple pathologic conditions such as insulin resistance, type-2 diabetes mellitus (T2DM), and metabolic dysfunction, which may also contribute to heart failure (HF) and death [4]. The World Health Organization (WHO) defines obesity as a degree of excess weight associated with “abnormal or excessive fat accumulation that may impair health” [5]. Commonly known as fat, adipose tissue is a lipid-storing connective tissue mainly composed of adipocytes with other minor cell types such as fibroblasts, endothelial cells, macrophages, stromal cells, immune cells, and pre-adipocytes [6].

The adipose tissue is distributed in various anatomic sites and is divided into specific regional depots that are different in cellular structural organization and biological function [7,8]. Among adipose tissues of different anatomic sites, epicardial adipose tissue (EAT) and pericardial adipose tissue (PAT) are most likely the principal adipose tissues relevant to the development of CVDs due to their close location to the myocardium. Nevertheless, according to currently available evidence, EAT may be more relevant than PAT for the following reasons. First, although they are both situated in the vicinity of the heart and share the same microcirculation, their anatomical locations are clearly different in detail. More specifically, EAT is located between the myocardium and the visceral pericardium, while PAT is located between the visceral pericardium and the parietal pericardium [9]. Thus, EAT is physically in contact with the myocardium, while PAT is physically separated from the myocardium by the visceral pericardium (Figure 1). Not only do they differ in location, but their biochemical function is different as well. It has been demonstrated that EAT is a metabolically active endocrine organ that secretes a number of adipokines and that interacts with the myocardium via a paracrine or vasocrine manner [10,11]. On the other hand, although there are few reports suggesting possible involvement of PAT in cardiometabolic processes [12,13], whether the PAT functions as an active endocrine entity is yet to be further elucidated.

In humans, two functionally distinct types of adipose tissues, namely white adipose tissue (WAT) and brown adipose tissue (BAT) [14], exist, and EAT falls into the category of an organ-enveloping, visceral type of WAT [8]. WAT primarily functions as fat storage to store excessive energy as triglycerides, and it releases various adipokines, hormones, and cytokines that regulate whole-body metabolism through paracrine and endocrine effects [15]. As a visceral type of WAT in the heart, EAT surrounds the majority of arteries as well as the heart, covering 80% of the heart’s surface and claiming 20% of the total heart weight [16]. Furthermore, and more importantly, it has been associated with the progression of cardiac dysfunction in obese individuals and the development of CVDs [17,18,19]. On the other hand, it has been also been reported that CVDs such as myocardial infarction (MI) stimulate EAT formation via the insulin-like growth factor 1 receptor (IGF1R)-dependent signaling pathway, suggesting a correlation between CVD progression and EAT [20]. As one of the mechanisms with which EAT contributes to the development of CVDs, various factors secreted from EAT such as adipokines, adrenomedullin, adiponectin, and micro-ribonucleic acids (microRNAs or miRNAs) have been identified [21].

MicroRNAs refer to relatively short (about 20 nucleotides long), non-coding RNAs that bind to target messenger RNAs (mRNAs) for degradation and/or translational repression, acting as a post-transcriptional regulator of the target genes [22]. Newly synthesized double-stranded mature miRNA is promptly separated into two strands, and a more stable one associates with the argonaute (Ago) protein, which is a component of the RNA-induced silencing complex (RISC) [23]. RISC-associated miRNA functions as a template to recognize the complementary sequence in the 3′ untranslated region (UTR) of target mRNAs [24]. As a result, RISC-miRNA complex-mediated gene silencing is achieved either by hindering the target mRNA translation or by degrading the target mRNA [13]. During the last few decades, this new class of small, non-coding RNAs has gained considerable recognition as a key regulator of virtually every cellular processes [25]. More than 60% of mammalian mRNAs are expected to be targeted by miRNAs, with a single mRNA that can be targeted by multiple miRNAs and vice versa [26]. Such unique features of miRNAs indicate that they are important regulators that fine tune the expression of hundreds of proteins [27]. Since the first discovery in 1993 [14], miRNAs have been implicated in various diseases, including CVDs [15,16,17]. For example, hyperglycemia, one of the well-known clinical pieces of evidence of obesity and T2DM, has been reported to alter the expression of EAT-derived miRNAs in a porcine model of obesity [28]. Therefore, it is not a far-fetched assumption that the pathophysiology of EAT during the development and progression of CVDs is regulated by miRNAs to a certain extent.

In the present review, we aim to summarize miRNAs reported to be differentially expressed in human EAT under pathologic conditions, focusing on their potential targets and their roles in the regulation of EAT biology. To carry out this narrative review, first, a PubMed search using “epicardial adipose tissue” and “miRNA” as keywords was conducted, which returned 36 articles. Among these, only those articles that either profiled the differential expression of EAT-derived miRNAs in patients with CVDs or investigated the molecular mechanism of a certain human EAT-derived miRNA were selected. The differentially expressed EAT-derived miRNAs identified in the profiling studies were further discussed for their potential targets and functions. Through this review, we hope to provide an idea of how the miRNA-dependent pathophysiological regulation of EAT can contribute to the development and progression of CVDs, and we hope to offer constructive suggestions for conducting future studies to investigate the role of miRNA-mediated EAT regulation in CVDs.

## 2. MicroRNAs in WAT Differentiation and Function

During the last decade, miRNAs have been demonstrated to target many of the key factors involved in the regulation of the different stages of adipogenesis, and numerous miRNAs whose expression changes during adipocyte differentiation have been identified. Although the underlying mechanisms by which these miRNAs regulate adipogenesis have not been fully elucidated, it is out of the question that miRNAs play a critical role in the regulation of adipogenesis by directly targeting key factors involved in adipocyte differentiation. Thus, some of the miRNAs reported to regulate the differentiation and function of WAT will be indicated and discussed here.

### 2.1. WAT Differentiation Inhibiting miRNAs

Normal physiologic regulation of WAT is critical for maintaining energy homeostasis because functionally impaired WAT cannot properly take up and store circulating lipids, which in turn, causes the accumulation of lipids in non-adipose tissue promoting various pathologic conditions, such as T2DM and atherosclerosis [29]. Differentiation of WAT from mesenchymal stem cells is achieved through multiple stages such as cell fate determination to generate pre-adipocytes, clonal expansion of pre-adipocytes, and terminal differentiation of pre-adipocytes into mature adipocytes. Previous studies have provided evidence that miRNAs play a critical role in the terminal differentiation of mature adipocytes and function. First, it was demonstrated that a number of adipogenesis-regulating miRNAs are dysregulated in WAT of human subjects with obesity and metabolic syndrome [30,31,32]. In addition, disruption of the key regulators of miRNA biosynthesis, such as Drosha and Dicer [33,34], is known to inhibit adipogenesis [35,36]. Especially, adipocyte-specific knockout of Dicer significantly decreased lipogenesis-related gene expression and depleted WAT, while it did not affect the lipogenesis of BAT, suggesting that miRNA-mediated regulation is more important in WAT than in BAT [37]. Since the inhibition of Drosha or Dicer has a global impact on overall miRNA expression, these studies are considered as examples of non-specific miRNA-dependent regulation of adipose tissue. On the other hand, there are also studies that have demonstrated specific miRNA-dependent regulation of the terminal differentiation of pre-adipocytes to mature adipocytes.

During the terminal differentiation of adipocytes, the expression of genes responsible for adipocyte function increases, and peroxisome proliferator-activated receptor gamma (PPARγ) and CCAAT/enhancer-binding protein alpha/beta (C/EBPα/β) are classical examples of the key transcription factors dictating the expression of such genes [38,39]. Therefore, miRNAs both directly and indirectly targeting these key transcription factors have been implicated in the regulation of adipocyte differentiation. For example, miR-130 is known to target PPARγ [40], and the expression of both miR-130a and 130b decreases as pre-adipocytes differentiate into mature adipocytes [41]. Furthermore, exogenous delivery of miR-130a or 130b significantly suppressed the expression of PPARγ as well as the expression of mature adipocyte markers such as leptin and adiponectin [41], suggesting that the miR-130 family can inhibit terminal adipocyte differentiation by targeting PPARγ. Another well-known miRNA that impairs adipocyte differentiation by directly targeting PPARγ is miR-27 [42,43].

Similar to the case of miR-130, the expression of miR-27a and 27b was demonstrated to be decreased in adipocytes of obese mice, and both of them inhibited adipogenesis by targeting PPARγ [44]. In addition to targeting PPARγ, miR-27 targets adipogenesis-promoting factors such as prohibitin (PHB) and cAMP response element binding protein (CREB). Being regulated by leptin, PHB functions as a critical factor in the insulin-induced differentiation of adipocytes [45], and it has been demonstrated that miR-27 suppresses adipocyte differentiation by targeting PHB [46]. CREB belongs to the bZIP superfamily of transcription factors [47], and it has been shown to play a crucial role in adipocyte differentiation [48]. According to a previous study, suppression of CREB by miR-27 also resulted in significant inhibition of adipose differentiation [49]. As such, miR-27 simultaneously suppresses multiple targets within pathways leading to adipogenesis, and this clearly exemplifies how a single miRNA can effectively modulate a certain biological process, including adipogenesis. Another example of miRNA inhibiting adipogenesis by targeting CREB is miR-155. Both CREB and C/EBPβ were found to be the target of miR-155, and miR-155-mediated suppression of those transcription factors was suggested to be the underlying mechanism of tumor necrosis factor alpha (TNF-α)-mediated inhibition of adipogenesis [50].

### 2.2. WAT Differentiation Promoting miRNAs

There are a number of miRNAs that promote adipogenesis by targeting factors and pathways known to suppress adipocyte differentiation. One of such miRNAs is the miR-30 family. The miR-30 family was demonstrated to be upregulated during adipogenic differentiation, and overexpression of miR-30a and miR-30d stimulated adipogenesis by targeting the osteogenic transcription factor, runt-related transcription factor 2 (RUNX2) [51]. Additional miRNAs targeting RUNX2 such as miR-204 [52] and -320 [53] have also been reported to increase adipogenesis, and miR-637 has been shown to increase adipogenesis by targeting another osteogenic transcription factor, Osterix [54]. Yet another member of the miR-30 family, miR-30c, has been reported to promote adipocyte differentiation by targeting plasminogen activator inhibitor-1 (PAI-1) and activin receptor-like kinase-2 (ALK2) [55]. ALK2 is a receptor of bone morphogenetic protein (BMP) that plays an important role in osteogenic differentiation [56]. Additionally, suppression of PAI-1 is known to increase the expression of PPARγ [57]. Therefore, the mechanism of miR-30c-stimulated adipogenesis seems to involve both suppression of osteogenesis and stimulation of adipogenesis. Considering the fate determination of mesenchymal stem cells (MSCs) depends on a fine tuning of adipo-osteogenic balance [58], and downregulation of master osteogenic transcription factors seems to be a reasonable approach for MSCs to enhance adipogenesis [59].

Both TNF-α [60] and transforming growth factor beta (TGF-β) [61] are known to be a potent inhibitor of adipocyte differentiation. Consequently, it can be speculated that miRNAs targeting factors constituting the TNF-α or TGF-β pathways can promote adipogenesis by counterbalancing the inhibitory effect during adipogenesis. Backing up such speculation, it has been demonstrated that miR-181a promotes adipogenesis by suppressing TNF-α expression [62], and miR-21 that targets TGF-β receptor type II (TGFBR2) also promotes adipogenesis [63]. Another signaling pathway known to suppress adipogenesis is Wingless-type MMTV integration site family (Wnt)-mediated pathway, which does so by downregulating PPAR and C/EBP [64]. Formerly known as transcription factor 4 (Tcf4), transcription factor 7, similar to 2 (Tcf7l2), is a key transcriptional effector downstream of the Wnt/β-catenin signaling pathway, and its polymorphism has been associated with the risk of T2DM [65]. Since Wnt signaling has a negative impact on adipogenesis, it has been reported that miRNAs boosting Wnt signaling by targeting glycogen synthase kinase 3 beta (GSK3β), an endogenous negative regulator of Wnt signaling [66], impairs adipogenesis [67,68]. Additional miRNAs promoting adipogenesis by targeting Wnt signaling include miR-148 and miR-210, and they are known to suppress adipogenesis by targeting Wnt signaling ligand WNT1 [69] and Tcf7l2 [70], respectively.

### 2.3. MicroRNAs Regulating WAT Functions

There are miRNAs known to regulate the functional aspects of WAT. Considering that diabetes contributes to the development of various cardiovascular diseases [71], regulation of insulin sensitivity may be one of the most relevant functions of WAT in terms of the risk for cardiovascular disease. To date, a number of miRNAs have been identified that regulate the WAT response to insulin, and a good example is miR-222. Regarding key regulators of insulin resistance, it has been reported that glucose transporter type 4 (GLUT4) levels in adipose tissue affect whole-body glucose homeostasis [72], as evidenced by the fat-specific knockout of GLUT4 resulting in insulin resistance in muscles and the liver [73]. Estrogens are other well-known regulators of insulin resistance, and they enhance the expression of GLUT4 via activation of estrogen receptor 1 (ESR1) [74]. MicroRNA 222 negatively regulates ESR1 [75], and its expression has increased in diabetic mice and as well as in the plasma of obese human [76,77]. Additionally, miR-222 is known to disrupt fatty acid metabolism by targeting acyl-CoA synthetase long-chain family member 4 (ACSL4) [78]. Therefore, miR-222 disrupts normal function of WAT by targeting multiple targets that facilitate adipocyte metabolism. Another example of miRNAs affecting WAT function is miR-369, which has been reported to target fatty acid binding protein 4 (FABP4) [79].

For normal physiologic insulin sensitivity, insulin receptor (IR) binds to caveolin-1 (CAV-1) activating CAV-1, and this initiates a series of auto-phosphorylation, leading to the translocation of GLUT4 to the plasma membrane surface where GLUT4 facilitates glucose uptake [80]. Therefore, depletion or suppression of CAV-1 impairs insulin-stimulated glucose uptake, leading to insulin resistance. This CAV-1 targeting mechanism of miRNAs producing a negative impact on insulin sensitivity has been also observed in the cases of miR-103 and-107 [81]. Similarly, miR-29 has been demonstrated to downregulate glucose uptake and GLUT4 expression by directly targeting secreted protein acidic rich in cysteine (SPARC) [82], whose dysregulation has been associated with obesity-related disorders including T2DM [83].

As a brief overview of how miRNAs modulate the function of WAT, a few examples have been indicated in this section (summarized in Table 1). Besides the above-mentioned miRNAs, there are many more miRNAs regulating the function of WAT [84]. However, for the purpose of this particular review, the role of more specific types of miRNAs, the EAT-derived miRNAs, in the development and progression of CVDs will be covered in detail in the following sections.

## 3. MicroRNAs Expressed in Human EAT

To date, a number of studies have examined the expression of EAT-derived miRNAs in relation to CVDs [85,86,87,88,89,90,91]. However, it is practically impossible to review all the EAT-derived miRNAs differentially expressed during the course of CVDs because a sizable portion of them have not been empirically examined pertaining to their possible impact on the cardiovascular system. Therefore, the main subject of this particular review will be limited to the EAT-derived miRNAs either whose effects on the cardiovascular system have been empirically proven or speculated to have an impact on the cardiovascular system based on published literature studies (Table 2).

According to a previous study conducted on integrative miRNA analyses of the EAT in patients with coronary artery disease (CAD), 15 miRNAs were significantly upregulated, while 14 miRNAs were downregulated in the EAT of CAD patients [85]. Among them, miR-135b-3p was the most significantly increased miRNA, and recent studies have investigated its role in the cardiovascular system. First, it has been reported to inhibit the proliferation, migration, and tube formation of endothelial cells (ECs) by targeting huntingtin-interacting protein 1 (HIP1) [92]. Therefore, increased expression of miR-135b-3p can negatively impact the tissue repair process following cardiac tissue damage. Another possible mechanism with which miR-135b-3p exerts negative impacts on the cardiovascular system is ferroptosis. Ferroptosis, an iron-dependent form of regulated cell death (RCD) [93], and the failure of the membrane lipid repair enzyme glutathione peroxidase 4 (GPX4) can induce accumulation of lipid peroxidase and reactive oxygen species (ROS), thereby inducing ferroptosis [94]. A recent study demonstrated that miR-135b-3p promoted ferroptosis of cardiomyocytes by targeting GPX4 [95].

Another miRNA significantly increased in the EAT of CAD patients was miR-1231 [85]. Voltage-dependent calcium channel subunit alpha2delta2 (CACNA2D) is known to be responsible for regulating calcium transportation and signal responses in cardiomyocytes [96], and it has been reported that miR-1231 aggravates arrhythmia by targeting CACNA2D2 [97]. Considering that the dysfunction of calcium channels has been closely associated with cardiac hypertrophy, arrhythmias, or even heart failure [98], it is likely that the increase in miR-1231 in EAT imposes potential harm to the heart.

MicroRNA-327-3p was another EAT-derived miRNA increased in CAD patients [85], and it is involved in the development of diabetic cardiomyopathy (DCM). Recently, it was reported that the downregulation of miR-372-3p promotes angiogenesis while suppressing oxidative stress, presumably by targeting phosphatidylinositol-4,5-bisphosphate 3-kinase catalytic subunit alpha (PIK3CA) in DCM mice [99]. That is to say that miR-372-3p can negatively regulate angiogenesis and increase oxidative stress. Consequently, increased levels of miR-372-3p may further aggravate CAD if there are additional underlying metabolic conditions such as diabetes. Additionally, regarding the inflammatory gene expression in EAT, miR-6870-3p has been positively linked to the elevated expression of inflammatory genes such as toll-like receptor 4 (TLR4), IL-6, c-Jun N-terminal kinases (JNK), nuclear factor kappa-light-chain-enhancer of activated B cells (NF-κB), and TNF-α in the EAT of CAD patients by targeting toll-interacting protein (TOLLIP) [100].

The miR-34 family, especially miR-34a, has been reported to be increased following MI in a rat model causing myocardial apoptosis and fibrosis by targeting the Wnt/β-catenin pathway [101]. miR-34a was one of the upregulated miRNAs in the EAT of the victims of sudden cardiac death (SCD) [91]. Considering that miRNAs can be encapsulated in biological vesicles such as exosomes and microvesicles, and transported out of the cells where they were synthesized to affect other cells and tissues [102], it is highly likely that even miRNAs synthesized in EAT can affect other type of cells residing in the heart, regulating their functions. Such encapsulated miRNAs derived from EAT will be covered in detail in the next section.

By their nature, miRNAs exert biological effects by downregulating the expression of their respective target genes. Therefore, if a certain miRNA targets genes considered cardioprotective, downregulation of such miRNA will have a negative impact on the cardiovascular system, and miR-193b-3p is one of such miRNAs. Adiponectin, a cytokine produced by adipocytes, is known to be anti-inflammatory, anti-atherogenic, and insulin-sensitizing and thus is considered cardioprotective [103]. One of the negative regulators of adiponectin production is nuclear transcription factor Yα (NF-YA) [104], and it has been reported that miR-193b-3p increases the expression of adiponectin by targeting NF-YA [105]. Therefore, downregulation of miR-193b-3p in the EAT of CAD patients [85] can lead to a decreased production of adiponectin, contributing to the development of cardiovascular complications. Another example of such miRNA whose downregulation may have a negative impact on the cardiovascular system is miR-3614.

In a recent study examining the differential expression of miRNAs in the EAT of CAD patients, it was demonstrated that the expression of miR-3614 was significantly decreased, and tumor necrosis factor receptor-associated factor 6 (TRAF6) was identified as one of the targets of miR-3614 [106]. TRAF6 is a key adapter protein that plays an important role in the induction of nuclear factor kappa-light-chain-enhancer of activated B cells (NF-κB)-dependent inflammatory response [107]. Therefore, downregulation of miR-3614, which targets TRAF6, may have a negative impact on the cardiovascular system by depriving means to prevent aberrant NF-κB signaling and subsequent overwhelming inflammatory response.

BAT is another example of biological entities that are considered cardioprotective [108]. As a thermogenic organ promoting glucose and lipid metabolism, BAT can enhance energy expenditure, and it can attenuate cardiac remodeling and suppress immune response by secreting adipokines such as fibroblast growth factor 21 (FGF21) [109] and IL-6 [110]. Furthermore, it has been suggested that EAT may function like BAT based on enhanced expression of BAT markers such as uncoupling protein 1 (UCP-1), PR domain containing 16 (PRDM16), and peroxisome proliferator-activated receptor gamma coactivator 1-alpha (PGC-1α) in EAT, and it can defend the myocardium and cardiac vessels against cardiac insults, including ischemia and/or hypoxia [111]. Therefore, by the same reasoning applied to miR-193b-3p, downregulation of a factor that promotes brown adipocyte differentiation will have a negative impact on the cardiovascular system, and miR-455-3p fits the profile. Bone morphogenetic protein 7 (BMP7) is a potent inducer of brown adipogenesis in multipotent mesenchymal cells [112], and as a downstream effector of BMP7, miR-455-3p induces PPARγ expression by suppressing Necdin, Runx1t1, and HIF1an, leading to the enhanced expression of BAT specific genes [113]. Thus, considering the powerful anti-inflammatory effect of BAT in the cardiovascular system, decreased expression of miR-455-3p in the EAT of CAD patients [85] may exacerbate the progression of CAD by not being able to provide sufficient cardiovascular protection that would have been achieved with normal, physiological expression level of miR-455-3p.

Such miRNA imposing negative impacts on the cardiovascular system has been reported in the EAT of CAD patients with T2DM. The expression of miR-146b-5p has been reported to be upregulated in the EAT of CAD patients with T2DM [87], and its role as a hypoxia-induced regulator of pro-fibrotic phenotype transition of cardiac cells has been demonstrated [114]. As the underlying mechanisms, suppressed expressions of interleukin 1 receptor associated kinase 1 (IRAK1), a key downstream effector of IL-1β signaling, and carcinoembryonic antigen related cell adhesion molecule 1 (CEACAM1), a cell–cell adhesion molecule, were identified.

In a study examining the role of miRNAs differentially expressed in the EAT of coronary atherosclerotic disease, the expression of miR-200 family members (miR-200b/c-3p, miR-141-3p and miR-429) was found to be significantly increased, and among theses, miR-200b-3p increased endothelial cell apoptosis by targeting histone deacetylase 4 (HDAC4) that is known to suppress endoplasmic reticulum (ER) stress-induced apoptosis by interacting with activating transcription factor 4 (ATF4) [115]. Additionally, that particular study demonstrated that a number of pro-inflammatory miRNAs, including miR-146a-5p, miR155-3p, miR-206 and miR-223-5p, were upregulated, while anti-inflammatory miRNAs such as Let-7i and miR-127-5p were downregulated. These findings suggested that the pro-inflammatory milieu of EAT could exacerbate the progression of atherosclerosis of the coronary arteries.

The positive correlation between cardiovascular disease and EAT-derived miRNAs also has been suggested in atrial fibrillation (AF) [116]. In this study, where the EAT area surrounding the left atrium (LA) was indexed to body surface area to obtain indexed LA EAT (iLAEAT), the authors reported that the bigger the iLAEAT is, the higher the chance of having hypertension, heart failure, and thick posterior walls. Furthermore, the expression of EAT-derived miRNAs, namely miRNAs 155-5p and 302a-3p, was significantly associated with iLAEAT in the patients with AF. Although it was not empirically proven, regulation of IL-8 and nerve growth factor (NGF) signaling was suggested as the underlying mechanism based on a pathway analysis. Since IL-8 has been associated with AF [117] and high plasma NGF can increase the incidence and duration of AF [118], upregulation of these factors may potentiate AF by creating atrial substrate for AF. Of note, miR-155 has been associated with other cardiovascular diseases such as CAD, abdominal aortic aneurysm (AAA), heart failure (HF), and diabetic heart disease (DHD) [119].

On the other hand, there is another miRNA that may protect the cardiovascular system by disrupting interleukin-1 (IL-1) signaling, one of the major inflammatory mechanisms of CVDs such as pericarditis, myocarditis, myocardial infarction, and heart failure [120]. MicroRNA-539-5p, one of the EAT-derived miRNAs increased in CAD patients [85], has been reported to target interleukin 1 receptor associated kinase 3 (IRAK3) and thereby inhibit inflammatory injury in cardiomyocytes [121]. One more interesting role of miR-539-5p is that it inhibits lipogenesis in WAT by targeting PPARγ [122]. Now, it is the general consensus that obesity predisposes to a pro-inflammatory state with increased inflammatory mediators such as IL-6 and tumor necrosis factor alpha (TNF-α) [123]. Thus, it is plausible that if the EAT in CAD patients creates enough inflammatory stress, as a protective response, the expression of miR-539-5p increases to suppress inflammatory signaling and further adipogenesis.

Similarly, miR-135b is one of the increased miRNAs in the EAT of CAD patients [85]. Since miR-135b and IL-1 receptor 1 (IL-1R1) form a negative feedback loop to resolve inflammation [124], such an increase in miR-135b may serve a protective role in EAT and the heart. Another example of such miRNA is miR-574-3p. It has been reported that the expression of miR-574-3p was increased in the EAT of CAD patients with T2DM [87]. According to a recent study examining the role of miR-574 in pathological cardiac remodeling, it suppressed the expression of the family with sequence similarity 210 member A (FAM210A), which interacts with mitochondrial translation elongation factor EF-Tu. Since the interaction of FAM210A with EF-Tu promotes mitochondrial-encoded protein expression, it was suggested that miR-574 maintains mitochondrial protein homeostasis and normal mitochondrial functions by preventing excessive expression of mitochondrial-encoded proteins in the electron transport chain [125].
biology-12-00498-t002_Table 2Table 2MicroRNAs reported to be differentially expressed in EAT of CVD patients.MicroRNAUnderlying CVD [Ref.]Expression ChangeReported Target, Effects Relevant to Cardiovascular SystemReferencemiR-135b^1^ CAD [85]Increased^2^ HIP1, inhibited ^3^ EC proliferation, migration, and tube formation[92]


^4^ GPX4, promoted ferroptosis of ^5^ CM[95]miR-1231CAD [85]Increased^6^ CACNA2D2, aggravated arrhythmia[97]miR-372-3pCAD [85]Increased^7^ PIK3CA, downregulation in ^8^ DCM mice promoted angiogenesis while suppressing oxidative stress[99]miR-6870-3pCAD [100]Increased^9^ TOLLIP, increased the expression of inflammatory genes (^10^ TLR4, ^11^ IL-6, ^12^ JNK, ^13^ NF-κB, and ^14^ TNF-α)[100]miR-34a^15^ SCD [91]Increased* Wnt1 and β-catenin, increased myocardial apoptosis and fibrosis following ^16^ MI in rats[101]miR-193b-3pCAD [85]Decreased^17^ NF-YA, increased the expression of adiponectin[105]miR-3614CAD [106]Decreased^18^ TRAF6, reduced ^19^ LPS-induced inflammatory injury[106]miR-455-3pCAD [85]DecreasedNecdin, ^20^ Runx1t1, and ^21^ HIF1an, promoted ^22^ BAT maturation[113]miR-146-5pCAD with ^23^ T2DM [87]Increased^24^ IRAK1 and ^25^ CEACAM1, promoted pro-fibrotic transition of cardiac cells[114]miR-200b-3pCAD [115]Increased^26^ HDAC4, increased EC apoptosis[115]miR-539-5pCAD [85]Increased^27^ IRAK3, inhibited inflammatory injury in CM [121]


^28^ PPARγ, inhibited ^29^ WAT lipogenesis[122]miR-135bCAD [85]Increased^30^ IL-1R1, forming a negative feedback loop with IL-1R1 to resolve inflammation[124]miR-574-3pCAD with T2DM [87]Increased^31^ FAM210A, maintained normal mitochondrial function [125]^1^ CAD: coronary artery disease, ^2^ HIP1: huntingtin-interacting protein 1, ^3^ EC: endothelial cell, ^4^ GPX4: glutathione peroxidase, ^5^ CM: cardiomyocytes, ^6^ CACNA2D2: voltage-dependent calcium channel subunit alpha2 delta2, ^7^ PIK3CA: phosphatidylinositol-4,5-bisphosphate 3-kinase catalytic subunit alpha, ^8^ DCM: diabetic cardiomyopathy, ^9^ TOLLIP: toll-interacting protein, ^10^ TLR4: toll like receptor 4, ^11^ IL-6: interleukin 6, ^12^ JNK: c-Jun N-terminal kinases, ^13^ NF-κB: nuclear factor kappa-light-chain-enhancer of activated B cells, ^14^ TNF-α: tumor necrosis factor alpha, ^15^ SCD: sudden cardiac death, ^16^ MI: myocardial infarction, ^17^ NF-YA: nuclear transcription factor Yα, ^18^ TRAF6: tumor necrosis factor receptor-associated factor 6, ^19^ LPS: lipopolysaccharide, ^20^ Runx1t1: RUNX1 partner transcriptional co-repressor 1, ^21^ HIF1an: hypoxia-inducible factor 1 alpha inhibitor, ^22^ BAT: brown adipose tissue, ^23^ T2DM: type 2 diabetes mellitus, ^24^ IRAK1: interleukin 1 receptor associated kinase 1, ^25^ CEACAM1: carcinoembryonic antigen related cell adhesion molecule 1, ^26^ HDAC4: histone deacetylase 4, ^27^ IRAK3: interleukin 1 receptor associated kinase 3, ^28^ PPARγ: peroxisome proliferator-activated receptor, ^29^ WAT: white adipose tissue, ^30^ IL-1R1: interleukin 1 receptor 1, ^31^ FAM210A: family with sequence similarity 210 member A. * Wnt1 and β-catenin expression was recovered by anti-miR-34a, however, no empirical validation of direct targeting was performed.


### Human EAT-Derived Exosomal miRNAs

Adipose tissue is one of the main sources of circulating exosomal miRNAs that affects the gene expression of other tissues [126]. Not only do the miRNAs of EAT play an important role in regulating glucose and lipid metabolism of the EAT itself, but they also mediate crosstalk between the EAT and the surrounding tissues or organs in the form of circulating miRNAs [127]. However, possibly due to limited availability of adequate samples for characterization of the EAT-derived exosomal miRNAs in the patients with CVDs, not that many studies on human EAT-derived exosomal miRNAs have been conducted, and it was only recently that such studies were published.

According to one of such studies that profiled EAT-derived exosomal miRNAs in CAD patients, 53 EAT-specific miRNAs were differentially expressed (adjusted *p* < 0.05, fold change > 2), of which 32 miRNAs were upregulated, while 21 of them were downregulated [128]. Among these, the increases in miR-141-3p, miR-183-5p, miR-200a-5p, miR-205-5p, and miR-429, as well as the decreases in miR-382-5p and miR-485-3p were further validated by quantitative polymerase chain reaction (qPCR). However, it is highly unlikely that a certain EAT-derived exosomal miRNA exerts significant biological effects on neighboring cardiac cells by not being sufficiently expressed under pathologic conditions, unless EAT is the one and only source of that specific miRNA throughout the whole body. Therefore, those EAT-derived exosomal miRNAs whose expression decreased in CAD patients are not covered in this review (Table 3).

Regarding the role of miR-141, it has been reported that miR-141 inhibited oxidized low-density lipoprotein (ox-LDL)-induced vascular smooth muscle cell (VSMC) proliferation by targeting PPARα [129]. Since ox-LDL can induce abnormal proliferation and migration of VSMCs [130] and contribute to the atherosclerotic plaque formation [131], upregulation of miR-141 may have an anti-atherosclerotic effect. Yet another study demonstrated that long noncoding RNA (lncRNA) metastasis-associated lung adenocarcinoma transcript-1 (MALAT1)-mediated downregulation of miR-141-3p/miR-200a-3p, which targets pro-atherosclerotic factor coiled-coil domain containing 80 (CCDC80), accelerated the development of atherosclerosis, also suggesting an anti-atherosclerotic role of miR-141 [132].

For the case of miR-138-5p, it has been reported that miR-138 induced human cardiomyocyte apoptosis by targeting sirtuin 1 (SIRT1), which negatively regulates p53 signaling by deacetylation [133]. On the other hand, there are also studies suggesting a cardioprotective role of miR-138. One of the targets of miR-138-5p is aldosterone synthase (CYP11B2), and miR-138 is known to reverse cardiac fibrotic remodeling during AF by suppressing CYP11B2 [134]. Similarly, in high glucose-induced cardiomyocyte damage, downregulation of miR-138-5p by its competitive endogenous RNA (ceRNA), lncRNA growth arrest-specific transcript 5 (GAS5), resulted in cardiomyocyte injury and apoptosis [135], suggesting its cardioprotective role. Furthermore, the observation that miR-138 attenuated hypoxia-induced cardiac cell death by targeting pyruvate dehydrogenase kinase 1 (PDK1) supports the cardioprotective effect of miR-138-5p [136].

Another miRNA reported to have both negative and positive impacts on the cardiovascular system is miR-200a-5p. As a negative impact, it has been reported that miR-200a-5p promoted selenium deficiency-induced myocardial necrosis by targeting ring finger protein 11 (RNF11) [137], which is known to negatively regulate NF-κB signaling [138]. Furthermore, overexpression of miR-200a-5p induced cardiomyocyte hypertrophy by upregulating glucose uptake while downregulating selenoprotein P (Sepp1), n (Seln), t (Selt), and 15 (Sep15) expression [139]. However, direct targeting interactions between these genes and miR-200a-5p has not been empirically validated. Additionally, it was reported that, by targeting cholesterol-transport gene LDL receptor related protein 1 (Lrp1) and ATP-binding cassette transporter subfamily A1 (Abca1), miR-205-5p promoted unstable atherosclerotic plaque formation [140]. However, there are also studies reporting the opposite.

It was reported that miR-200a protects cardiomyocytes from hypoxic insult, increasing cell survival while inhibiting apoptosis and reactive oxygen species (ROS) production [141]. In that particular study, miR-200-5p-mediated het downregulation of kelch-like ECH-associated protein 1 (Keap1), a negative regulator of nuclear factor erythroid-2-related factor 2 (Nrf2) [142], which was identified as an underlying mechanism. Such a cardioprotective effect of miR-200-5p by downregulating Keap1/Nrf2 axis was recently validated again in an independent study [143].

Another miRNA upregulated in EAT-derived exosomal miRNA is miR-205-5p, and its role in VSMC proliferation has been reported in two independent studies. The first study examined its role in pulmonary VSMC (pVSMC) proliferation and demonstrated that restoration of miR-205-5p, which was downregulated during hypoxia-induced pulmonary arterial hypertension (PAH), inhibited pVSMC proliferation by targeting molecules interacting with CasL 2 (MICAL2) [144], a cytoskeleton dynamics regulator known to promote cancer cell proliferation [145]. In the other study, receptor tyrosine protein kinase erbB 4 (ERBB4), known to promote phosphatidylinositol 3-kinase (PI3K)/Akt signaling and subsequent cell proliferation [146], was identified as one of the targets of miR-205-5p, and thereby, miR-205-5p inhibited ox-LDL induced VSMC proliferation and migration [147]. Although phenotype switching of VSMC from the quiescent contractile phenotype to a proliferative, migratory, and synthetic phenotype is one of the triggering factor in arteriogenesis required for tissue repair following ischemia [148], aberrant proliferation of VSMCs is more often linked to atherosclerosis [149]. Furthermore, since miR-205-5p in these particular studies inhibited VSMC proliferation induced by pathologic stimuli, it seems safe to state that miR-205-5p played a cardioprotective role in those particular studies.

Another example of cardioprotective effects of miR-205-5p was demonstrated in high-fat diet (HFD)-induced atrial fibrosis [150]. According to that particular study, miR-205-5p expression was downregulated in HFD-induced atrial fibrosis, and restoration of miR-205-5p improved atrial fibrosis, along with decreased mitochondrial damage, by targeting euchromatic histone-lysine N-methyltransferase 2 (EHMT2). The primary function of EHMT2 is to di-methylate lysine 9 of histone H3 (H3K9me2) [151], and hyper-methylation of insulin-like growth factor binding protein 3 (IGFBP3) promoter decreases IGFBP3 expression [152]. Therefore, downregulation of EHMT3 by miR-205-5p can enhance the expression of IGFBP3 by preventing hyper-methylation of the IGFBP3 promoter. Since the expression pattern of IGFBP3 in various cardiovascular diseases can vary depending on the underlying conditions [153], it is not accurate to simply state that increased IGFBP3 has cardioprotective effects. Nevertheless, considering that IGFBP3 exerts anti-inflammatory effects by suppressing the NF-κB signaling pathway [154] and that inflammation exacerbates HFD-induced cardiac fibrosis [155], it can be speculated that IGFBP3 improves HFD-induced cardiac fibrosis by suppressing inflammatory responses.

The last miRNA demonstrated to be upregulated in the exosomes derived from the EAT of CAD patients is miR-429. In relation to the cardiovascular system, it has been reported that miR-429 enhances pro-inflammatory responses in VSMCs of diabetic mice by targeting zinc finger E-box-binding homeobox 1 (Zeb1), an E-box binding transcriptional repressor of inflammatory genes such as monocyte chemoattractant protein 1 (MCP-1) and cyclooxygenase 2 (COX2) [156]. Furthermore, miR-429 induced endothelial cell apoptosis by targeting B-cell lymphoma 2 (Bcl2) under atherosclerotic conditions [157]. Given that exosomal miRNAs can be transferred and can exert their biological effect on neighboring cells, it is speculated that increased miR-429 may exacerbate CADs by increasing inflammatory responses and apoptosis in vascular cells of the heart.
biology-12-00498-t003_Table 3Table 3EAT-derived exosomal miRNAs increased in CVD patients.MicroRNAReported Target, Effects Relevant to Cardiovascular SystemReferencemiR-141^1^ PPARγ, inhibited ^2^ ox-LDL induced proliferation of VSMCs[129]^3^ CCDC80, ^4^ MALAT1-mediated downregulation of miR-141-3p promoted atherosclerosis[132]miR-183-5p^5^ SIRT1, induced ^6^ CM apoptosis[133]^7^ CYP11B2, reversed cardiac fibrotic remodeling[134]^8^ PDK1, attenuated hypoxia-induced cardiac cell death[136]miR-200a-5p^9^ RNF11, promoted selenium deficiency-induced myocardial necrosis[137]* Selenoprotein P, n, t, and 15, induced CM hypertrophy[139]^10^ Lrp1 and ^11^ Abca1, promoted unstable atherosclerotic plaque formation[140]^12^ Keap2, promoted pro-fibrotic transition of cardiac cells[141]miR-205-5p^13^ MICAL2, inhibited pulmonary ^14^ VSMC proliferation[144]^15^ ERBB4, inhibited ox-LDL induced VSMC proliferation and migration[147]^16^ EHMT2, improved ^17^ HFD-induced atrial fibrosis[150]miR-429^18^ Zeb1, promoted pro-inflammatory response in VSMC of diabetic mice[156]^19^ Bcl2, induced ^20^ EC apoptosis under atherosclerotic condition[157]^1^ PPARγ: peroxisome proliferator-activated receptor, ^2^ ox-LDL: oxidized low density lipoprotein, ^3^ CCDC80: coiled-coil domain containing 80, ^4^ MALAT1: metastasis-associated lung adenocarcinoma transcript 1, ^5^ SIRT1: sirtuin 1 ^6^ CM: cardiomyocytes, ^7^ CYP11B2: aldosterone synthase, ^8^ PDK1: pyruvate dehydrogenase kinase 1, ^9^ RNF11: ring finger protein 11, ^10^ Lrp1: low density lipoprotein receptor related protein 1, ^11^ Abca1: ATP-binding cassette transporter subfamily A1, ^12^ Keap1: kelch-like ECH-associated protein 1, ^13^ MICAL2: molecule interacting with CasL2, ^14^ VSMC: vascular smooth muscle cells, ^15^ ERBB4: receptor tyrosine protein kinase erbB 4, ^16^ EHMT2: euchromatic histone-lysine N-methyltransferase, ^17^ HFD: high fat diet, ^18^ Zeb1: zinc finger E-box-binding homeobox 1, ^19^ Bcl2: B-cell lymphoma 2, ^20^ EC: endothelial cells. * Direct targeting interactions between these genes and miR-200a-5p was not empirically validated.


## 4. Discussion

As discussed above, accumulating evidence indicates that a number of human EAT-derived miRNAs are constantly in action. Some are believed to be cardioprotective, while some others have demonstrated quite opposite effects on the cardiovascular system, depending on the underlying pathologic conditions such as CAD, MI, DCM, AF, and HF. Therefore, it is the fine balance between physiologic stimuli and pathologic stimuli that dictates the role of a given EAT-derived miRNA in the cardiovascular system. Additionally, due to the multi-targeting nature of miRNAs, as well as highly limited availability of human EAT samples, it is very difficult to make generalized claims on a given miRNA in terms of its overall impact on the cardiovascular system, and thus, many of the underlying mechanisms with which EAT-derived miRNAs regulate both physiology and pathophysiology of the cardiovascular system remain largely speculative. In that context, any mechanism of action discussed or speculated in this review can only serve as a reference for conducting future studies.

Accumulating evidence strongly suggests the involvement of EAT-derived miRNAs in the regulation of the cardiovascular system, and therefore, it is commonly acknowledged that EAT-derived miRNAs have great potential as both diagnostic and therapeutic modalities. Most of the studies covered by this review involved profiling of the EAT-derived miRNAs in patients with cardiovascular-related diseases. A profiling approach seems to be logical and sufficient for finding novel and reliable biomarkers of cardiovascular disease. However, it may not be sufficient for designating a certain miRNA as a therapeutic target because it might as well be a simple bystander or byproduct of an already progressed pathologic process rather than a key modulator involved in the initiation and/or progression of a given CVD. A more in-depth understanding of the biological characteristics of EAT-derived miRNAs will provide a vital theoretical basis for the prevention and treatment of disease, and such understanding necessitates functional investigation of a given miRNA, including, but not limited to, the study of its dose effect, off-target effects, and potential toxicity.

## 5. Conclusions

In the present review, human EAT-derived miRNAs reported to be differentially expressed under pathologic conditions are discussed and summarized, and we hope that this review can provide novel insights to transform our current knowledge on human EAT-derived miRNAs into clinically viable therapeutic strategies for preventing and treating CVDs. Considering the anatomical proximity of EAT to the heart and the existence of EAT-derived exosomal miRNAs, which show a wide spectrum of anti-inflammatory to pro-inflammatory effects, it is clear that EAT-derived miRNAs make a vital contribution to the cardiovascular system both locally and systemically, and a comprehensive understanding of their function and mechanisms may hold the key for the prevention and treatment of CVDs.

## Figures and Tables

**Figure 1 biology-12-00498-f001:**
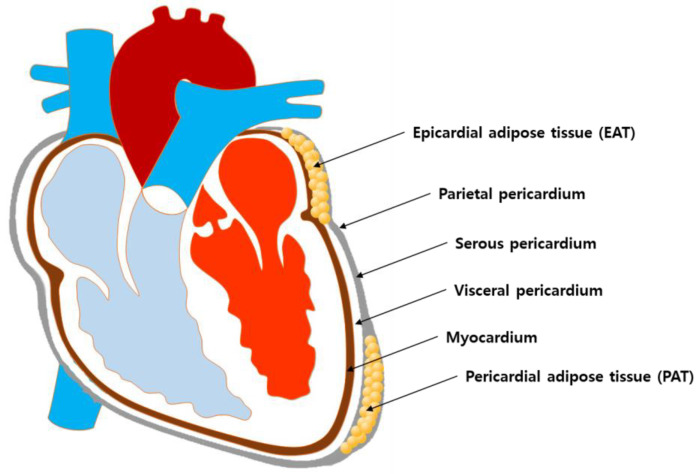
Schematics of the anatomical location of EAT and PAT in the heart.

**Table 1 biology-12-00498-t001:** Examples of miRNAs regulating terminal differentiation and function of WAT.

MicroRNA	Direct Target (Reported Effects, If Applicable)	Reference
**WAT Differentiation Inhibitory**		
miR-130a, miR-130b	^1^ PPARγ	[40,41]
miR-27a, miR-27b	PPARγ	[42,43,44]
	^2^ PHB	[46]
	^3^ CREB	[49]
miR-155	CREB and ^4^ C/EBPβ	[50]
**WAT Differentiation Promoting**		
miR-30a, miR-30d	^5^ RUNX2	[51]
miR-204	RUNX2	[52]
miR-320	RUNX2	[53]
miR-637	Osterix	[54]
miR-30c	^6^ PAI-1 and ^7^ ALK2	[55]
miR-181a	^8^ TNF-α	[62]
miR-21	^9^ TGFBR2	[63]
miR-148	^10^ WNT1	[69]
miR-210	^11^ Tcf7l2	[70]
**WAT Function Regulating**		
miR-222	^12^ ESR1 (insulin resistance)	[75]
	^13^ ACSL4 (fatty acid metabolism disruption)	[78]
miR-369	^14^ FABP4 (adipogenesis inhibition)	[79]
miR-103, miR-107	^15^ CAV-1 (disrupted glucose uptake, insulin resistance)	[81]
miR-29	^16^ SPARC (decreased glucose uptake)	[82]

^1^ PPARγ: peroxisome proliferator-activated receptor gamma, ^2^ PHB: prohibitin, ^3^ CREB: cAMP response element binding protein, ^4^ C/EBPβ: CCAAT/enhancer-binding protein beta, ^5^ RUNX2: runt-related transcription factor 2, ^6^ PAI-1: plasminogen activator inhibitor-1, ^7^ ALK2: activin receptor-like kinase-2, ^8^ TNF-α: tumor necrosis factor alpha, ^9^ TGFBR2: transforming growth factor beta receptor 2, ^10^ WNT1: proto-oncogene Wnt-1, ^11^ Tcf7l2: transcription factor 4, transcription factor 7 like 2, ^12^ ESR1: estrogen receptor 1, ^13^ ACSL4: acyl-CoA synthetase long-chain family member 4, ^14^ FABP4: fatty acid binding protein 4, ^15^ CAV-1: caveolin-1, ^16^ SPARC: secreted protein acidic rich in cysteine.

## Data Availability

Not applicable.

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
