# Peer review of "The Role of Epicardial Adipose Tissue-Derived MicroRNAs in the Regulation of Cardiovascular Disease: A Narrative Review"

_biology, 2023, doi:10.3390/biology12040498_

Round 1
Reviewer 1 Report
This article develops an extensive review focused on elucidating the role of miRNAs derived from epicardial adipose tissue in cardiovascular pathologies. As indicated in the conclusions, this is a great source of information to continue with the study in this field, with a view to improving diagnostic methods, also focusing on finding therapeutic targets for the treatment of this type of pathology.
It is necessary to respond to some minor observations.
The methodology used to carry out the review is not specified in the manuscript. It is recommended to include this information to give even more quality to the article. On the other hand, it is necessary to include in the title what type of review was carried out. Probably, in this case a narrative review of the literature was carried out (the inclusion and exclusion criteria and the methodology followed for the selection of the articles have not been specified). This will facilitate search strategies once the article has been published.
Best regards.
Author Response
This article develops an extensive review focused on elucidating the role of miRNAs derived from epicardial adipose tissue in cardiovascular pathologies. As indicated in the conclusions, this is a great source of information to continue with the study in this field, with a view to improving diagnostic methods, also focusing on finding therapeutic targets for the treatment of this type of pathology. It is necessary to respond to some minor observations.
The methodology used to carry out the review is not specified in the manuscript. It is recommended to include this information to give even more quality to the article.
Response: To comply with the reviewer's recommendation, the following sentences were added in the last paragraph of the revised introduction (line 114-120); “To carry out this narrative review, first, Pubmed search using “epicardial adipose tissue” and "miRNA" as keywords was conducted and it brought up 36 articles. Among these, only the articles either profiled the differential expression of EAT-derived miRNAs in patients with CVDs or investigated the molecular mechanism of a certain human EAT-derived miRNAs were selected. The differentially expressed EAT-derived miRNAs identified in the profiling studies were further discussed for their potential targets and functions.”
On the other hand, it is necessary to include in the title what type of review was carried out. Probably, in this case a narrative review of the literature was carried out (the inclusion and exclusion criteria and the methodology followed for the selection of the articles have not been specified). This will facilitate search strategies once the article has been published.
Response: Title has been modified with “:a narrative review” at the end to clearly indicate that this review is a narrative review as the reviewer requested.
Reviewer 2 Report
I have two concerns for this review.
1. A figure summarized targets for miRNA is required.
2. The authors have to be careful about the references on miRNA research from non-famous Chinese hospital and journals. Probably, some are written by paper mill.
Author Response
I have two concerns for this review.
- A figure summarized targets for miRNA is required.
Response: In fact, we also considered to create a figure summarizing EAT-derived miRNAs and their corresponding targets for the initial submission. However, we covered a total of 18 miRNAs and 37 corresponding targets relating to EAT in this review (this does not include 20 miRNAs and 17 targets involved in WAT differentiation and function), and we worried that it might be less informative if we put all the information in one figure because listing all the miRNAs, targets, and biological consequences in one figure may make it rather difficult to comprehend. Therefore, we instead created separate tables showing miRNAs, corresponding targets, and observed biological effects for both EAT expressed miRNAs (Table 2) and EAT-derived exosomal miRNAs (Table 3), as well as for the ones involved in WAT differentiation and function (Table 1). With all due respect, please accept our apologies and understand that we cannot fully comply with the reviewer's request on this issue.
- The authors have to be careful about the references on miRNA research from non-famous Chinese hospital and journals. Probably, some are written by paper mill.
Response: Thanks for the advice and we fully agree with the reviewer's concern on the papers produced by paper mills. However, it can be difficult to discern bad apples based on where they came from without a cold evidence because even a paper mill-produced paper can be published in apparently legitimate journals. Therefore, although we know that paper mill-produced papers are different from predatory journals & publishers, we checked whether any of our cited references is listed as potential predatory journals & publishers based on the Beall's list (https://beallslist.net/standalone-journals/) to minimize the risk of citing papers of poor quality. Fortunately, none of them were on the list this time, but we will always keep the reviewer's advice in our minds for future manuscripts as well.
Reviewer 3 Report
The authors have written a comprehensive review on the role of epicardial adipose tissue-derived microRNAs in cardiovascular disease. They have introduced the topic very soundly, providing a comprehensive background on the lineage and factors regulating the adipose tissue differentiation. And they are then focusing on the epicardial-derived miR. I don’t have any significant comments on the paper.
General comments:
There are some grammatical errors in the paper.
The authors have thoroughly included the relevant literature in their review. But, briefly adding the following references and discussing them could further strengthen the paper.
MicroRNA-3614 regulates inflammatory response via targeting TRAF6-mediated MAPKs and NF-κB signaling in the epicardial adipose tissue with coronary artery disease. PMID: 32950591
Hyperglycemia-induced changes in miRNA expression patterns in epicardial adipose tissue of piglets. PMID: 27044779
An IGF1R-dependent pathway drives epicardial adipose tissue formation after myocardial injury. PMID: 27803039
Author Response
The authors have written a comprehensive review on the role of epicardial adipose tissue-derived microRNAs in cardiovascular disease. They have introduced the topic very soundly, providing a comprehensive background on the lineage and factors regulating the adipose tissue differentiation. And they are then focusing on the epicardial-derived miR. I don’t have any significant comments on the paper.
General comments:
There are some grammatical errors in the paper.
Response: we carefully and thoroughly checked the manuscript for any errors again.
The authors have thoroughly included the relevant literature in their review. But, briefly adding the following references and discussing them could further strengthen the paper.
Suggested Ref. 1. MicroRNA-3614 regulates inflammatory response via targeting TRAF6-mediated MAPKs and NF-κB signaling in the epicardial adipose tissue with coronary artery disease. PMID: 32950591
Suggested Ref. 2. Hyperglycemia-induced changes in miRNA expression patterns in epicardial adipose tissue of piglets. PMID: 27044779
Suggested Ref. 3. An IGF1R-dependent pathway drives epicardial adipose tissue formation after myocardial injury. PMID: 27803039
Response: suggested references have been added to the manuscript where we think they fit without compromising the main theme of the review (Ref. 1: line 354-364, ref. 2: line 107-109, and ref 3: line 86-88).